# Newly Diagnosed Type 2 Diabetes Care between Family Physicians, Endocrinologists, and Other Internists in Taiwan: A Retrospective Population-Based Cohort Study

**DOI:** 10.3390/jpm12030461

**Published:** 2022-03-14

**Authors:** Pei-Lin Chou, I-Hui Chiang, Chi-Wei Lin, His-Hao Wang, Hao-Kuang Wang, Chi-Hsien Huang, Chao-Sung Chang, Ru-Yi Huang, Chung-Ying Lin

**Affiliations:** 1Department of Family and Community Medicine, E-Da Hospital, Kaohsiung 82445, Taiwan; ngilriw@gmail.com (P.-L.C.); ed109959@edah.org.tw (I.-H.C.); ed104283@edah.org.tw (C.-W.L.); ed103520@edah.org.tw (C.-H.H.); 2College of Medicine, I-Shou University, Kaohsiung 82445, Taiwan; ed103631@edah.org.tw (H.-H.W.); ed101393@edah.org.tw (H.-K.W.); ed107670@edah.org.tw (C.-S.C.); 3Division of Nephrology, Department of Internal Medicine, E-Da Hospital, Kaohsiung 82445, Taiwan; 4Department of Medical Quality, E-Da Hospital, Kaohsiung 82445, Taiwan; 5Department of Neurosurgery, E-Da Hospital, Kaohsiung 82445, Taiwan; 6Committee for Advanced Medical Technology, E-Da Hospital, Kaohsiung 82445, Taiwan; 7Institute of Allied Health Sciences, College of Medicine, National Cheng Kung University, Tainan 70101, Taiwan; 8Department of Occupational Therapy, College of Medicine, National Cheng Kung University, Tainan 70101, Taiwan; 9Biostatistics Consulting Center, National Cheng Kung University Hospital, College of Medicine, National Cheng Kung University, Tainan 70101, Taiwan; 10Department of Public Health, College of Medicine, National Cheng Kung University, Tainan 70101, Taiwan

**Keywords:** diabetes care, generalists, specialists, family physician, vascular complications

## Abstract

(1) Background: We aimed to determine whether physicians of different specialties perform differently in the monitoring, cost control, and prevention of acute outcomes in diabetes care. (2) Methods: Using data from the Health and Welfare Data Science Center, participants with newly diagnosed type 2 diabetes (*n* = 206,819) were classified into three cohorts based on their primary care physician during the first year of diagnosis: family medicine (FM), endocrinologist, and other internal medicine (IM). The three cohorts were matched in a pairwise manner (FM (*n* = 28,269) vs. IM (*n* = 28,269); FM (*n* = 23,407) vs. endocrinologist (*n* = 23,407); IM (*n* = 43,693) vs. endocrinologist (*n* = 43,693)) and evaluated for process indicators, expenditure on diabetes care, and incidence of acute complications (using subdistribution hazard ratio; sHR). (3) Results: Compared to the FM cohort, both the IM (sHR, 1.26; 95% CI, 1.08 to 1.47) and endocrinologist cohorts (sHR, 1.57; 95% CI, 1.38–1.78) had higher incidences of acute complications. The FM cohort incurred lower costs than the IM cohort (USD 487.41 vs. USD 507.67, *p* = 0.01) and expended less than half of the diabetes-related costs of the endocrinology cohort (USD 484.39 vs. USD 927.85, *p* < 0.001). (4) Conclusion: Family physicians may provide better care at a lower cost to newly diagnosed type 2 diabetes patients. Relatively higher costs incurred by other internists and endocrinologists in the process of diabetes care may be explained by the more frequent ordering of specialized tests.

## 1. Introduction

Uncontrolled diabetes can lead to numerous comorbid health problems that lower patients’ quality of life and inflict a significant economic burden. The overall diabetes-related burden is still high because of the growing prevalence of diabetes, which is projected to be increased. [1] High quality and cost-saving diabetes care is therefore essential on individual and societal levels. The National Committee on Quality Assurance (NCQA) and the American Diabetes Association (ADA) has developed a guideline for quality diabetes care, including measures such as periodic testing of glycohemoglobin (HbA1c), low-density lipoprotein (LDL) levels, and microalbuminuria in an effort to prevent negative outcomes such as acute and chronic vascular complications. [2,3] However, several studies have revealed that current diabetes care still has room for improvement [4,5,6] such as control of risk factors or achieving guideline-recommended targets, of which are usually the main services provided by family physicians to take care of their patients. In Taiwan, specifically, studies have shown suboptimal diabetes care leading to the emergence of vascular complications in patients [7,8,9].

The reason diabetes control is so commonly unsatisfactory is multifactorial. Based on a previous study, [10] a disease-centered medical approach, lack of knowledge, and limited patient participation in the decision-making process are major contributors. Another study by Shani et al. [11] found that a patient’s primary care physician is the most important predictor of quality of diabetes care. Physicians of different specialties may be more or less effective at managing diabetes patients based on differences in their training. For instance, it may be postulated that while endocrinologists have a deeper fund of diabetes knowledge, general and family physicians may have more patient-centered approaches in their practices, which may influence the quality of diabetes care given [12]. How the possible strengths and weaknesses of care provided by physicians of varying specialties translate to quantifiable outcome measures is an important question. Previous studies have focused on comparing endocrinologists with non-endocrinologists, and some have suggested that care by endocrinologists is associated with better monitoring of diabetes process measures, which include the frequency of laboratory measurements (e.g., HbA1c, lipids, and microalbuminuria), diabetic feet, and eye examinations [13,14]. Conversely, other studies have found that there were no statistical differences in any quality measures [15] or mortality when comparing care provided by different specialties [16].

Under Taiwan’s National Health Insurance (NHI) system, patients are permitted to seek care from physicians of any specialty without prior evaluation and referral by a primary care provider. Consequently, patients receive diabetes care from physicians of a range of specialties, with the most common being family physicians, endocrinologists, and other internists. However, the quality of diabetes care may differ between specialties in the process of updating knowledge on care and treatment modalities. Tseng et al. [17] reported that in Taiwan, patients with diabetes cared for by endocrinologists had better glucose control and adherence to treatment than those cared for by family physicians. Liu et al. [18] demonstrated that diabetes patients who were not consistently cared for by an endocrinologist had a significantly higher risk of hospitalization due to diabetic ketoacidosis (DKA). However, these data were obtained from a single center and mainly compared diabetes care provided only by endocrinologists versus physicians of other specialties. Moreover, previous studies did not investigate the health expenditure of diabetes care between different specialties. Therefore, we aimed to examine the differences in acute complications, costs, and the process of diabetes care given by family physicians, endocrinologists, and other internists using data from the Health and Welfare Data Science Center (HWDC).

## 2. Materials and Methods

### 2.1. Source of Data

The Taiwan Ministry of Health and Welfare shifted to a single-payer system under the NHI program in 1995 and today covers the healthcare of more than 99% of Taiwanese residents. The National Health Research Institute (NHRI) of Taiwan manages the medical benefits claims of all residents in Taiwan and has made several datasets available for public use. We requested information from the Data Science Center of the Taiwan Ministry of Health and Welfare, which covers claims data from 2000 to 2017 for data management and analyses [19]. The completeness and accuracy of the datasets were guaranteed by the Department of Health and the NHI Bureau of Taiwan. The HWDC contains the complete original claims data of approximately 23 million insured individuals with 70 other health-related databases for data management and analyses in 2000 and 2017. Until the end of 2015, all individuals were followed up for outcome identification using the International Classification of Disease, 9th Revision, and Clinical Modification (ICD-9-CM), then afterwards using the International Classification of Disease, 10th version, and Clinical Modification (ICD-10-CM) system.

### 2.2. Study Design and Study Population

We selected subjects from the HWDC aged 18–75 years between 2000 and 2014. These subjects were followed for one year to verify the diagnoses of new-onset type 2 diabetes mellitus and then followed for two more years for data collection of diabetes care quality. The index day was defined as when the patient was diagnosed with new-onset type 2 diabetes, which was one year after entering the study. The study endpoint was defined as the day of the diagnosis of acute complications, death, or until 31 December 2017. We collected information such as age, gender (male/female), level of urbanization (high/moderate/low), frequency of insulin usage during the first month of diagnosis, and comorbidities according to the Charlson comorbidity index (Appendix A) at baseline (i.e., the index date). Participants with new-onset type 2 diabetes were defined as those meeting at least one of the following inclusion criteria: [20,21] (1) one or more inpatient admissions with a diagnosis of diabetes mellitus; (2) four or more outpatient visits within a 1-year period, each with a diagnosis of diabetes mellitus; or (3) outpatients who were diagnosed with diabetes mellitus and prescribed diabetes drugs during the same visit. Participants were excluded if they were any of the following: (1) not new-onset type 2 diabetes (i.e., specifically excluded those with any diabetes mellitus diagnosis prior to 2000, or the first year of study entry), occurrence of chronic complications suspected to be secondary to diabetes (i.e., diabetic renal, ophthalmic, neurologic, and peripheral vascular disease) before the diagnosis of diabetes mellitus (*n* = 2662); (2) the primary care physician in the majority of clinical visits during the first and follow-up year was not a family physician, endocrinologist, or other internist (*n* = 841,782); (3) patients aged <18 or >75 years (*n* = 352,623); (4) patients who had no available data on sex or residence (sex *n* = 2090, area *n* = 33,458); (5) patients who died or had acute complications within 1 year of diagnosis (*n* = 58,007).

We divided the participants into three cohorts based on the type of physician they saw for more than half of their visits for diabetes treatment in the first year of diagnosis: family medicine (FM), endocrinologist, and other internal medicine (IM) cohorts. Specialties included in the other internal medicine cohort were cardiology, nephrology, general medicine, and neurology, which are the specialties besides endocrinology and family medicine most likely to be responsible for diabetes care in Taiwan. Next, the three cohorts were matched in pairs (i.e., FM versus IM cohort, FM versus endocrinologist cohort, and IM versus endocrinologist cohort) in a 1:1 ratio according to age, gender, index day, Charlson comorbidity index, and level of urbanization. The characteristics of participants who can not be categorized based on the primary care physician were presented as others in the Appendix A.

### 2.3. Outcome Indicators in Two-Year Follow-Up

The outcome indicators included (1) process indicators, (2) costs, and (3) acute complications. We extracted process indicators, including the following: frequency of testing for HbA1c, lipid profile, and urinary albumin-to-creatinine ratio (ACR) during the 2-year follow-up period. We further collected information regarding the average diabetes-related annual costs (converted to USD using the exchange rate in 2017). Fees of diagnosis, treatment, and medications represented the diabetes-related expenditures. The outcome indicators of this study were the incidence of acute complications during the follow-up period. Acute complications of interest were DKA, hyperosmolar hyperglycemic state (HHS), and hypoglycemia episodes noted in the medical records from all sources of medical service, i.e., outpatient, emergency, and inpatient department visits, during the 2-year follow-up period.

### 2.4. Statistical Analysis and Comorbidity Risk Analysis

Differences among groups (FM versus IM cohort, FM versus endocrinologist cohort, IM versus endocrinologist cohort) were evaluated using the t-test for continuous variables and the chi-square test for categorical variables. The study considered death as a competing event to calculate the risk of acute complications. We calculated the subdistribution hazard ratio (sHR) using the Fine and Gray regression hazards model, and *p*-values were determined using Gray’s test [22]. The univariate sHR and multivariate-adjusted sHR accounted for age, sex, tests within the 2-year follow-up period, and insulin use within the first month of diagnosis. Cost and comorbidities at baseline were expressed with the 95% confidence interval (CI) and the two-sided *p*-value. To determine the cumulative incidence of acute complications and survival probability, the Kaplan–Meier method was used in both analyses, and differences between cohorts were tested using the log-rank test. Statistical significance was set at *p* < 0.05. All data management and sHR calculations were conducted using the Statistical Analysis System software for Windows (version 9.4; SAS Institute, Cary, NC, USA). The Fine and Gray regression hazard model was performed using the PHREG package.

## 3. Results

### 3.1. Participants

Eligible participants from January 2000 to December 2014 were enrolled in the analysis (Figure 1). A total of 1,497,441 participants were enrolled at first, and after excluding ineligible participants, those who saw doctors of the same specialty for more than 50% of medical visits for diabetes in the first year were categorized into the FM cohort (*n* = 36,979), IM cohort (*n* = 99,116), or endocrinologist cohort (*n* = 70,724).

After matching, the FM and IM cohorts (28,269 vs. 28,269) had similar ages and overall comorbidities, but the IM cohort had a higher initial prescription of insulin by 0.15 times/month and a higher annual diabetes-related cost of USD 20. Furthermore, participants in the FM cohort received a slightly higher testing frequency for HbA1c and lipid profiles annually. On the other hand, those in the IM cohort had a higher testing frequency for ACR annually (Table 1).

The FM and endocrinologist cohorts were also matched (23,407 vs. 23,407) Patients under the care of endocrinologists had a significant 0.5 times more insulin prescriptions per month and almost spent twice as much on diabetes care. Notably, the participants in the endocrinologist cohort also received significantly higher annual testing frequencies for all types (Table 2).

When comparing the endocrinologist to IM cohorts (43,693 vs. 43,693), the endocrinologist cohort also had significantly higher frequency of all types of testing, prescription of insulin (0.4 times/month), and diabetes-related costs (USD 430 more) (Table 3).

### 3.2. Process Indicators

The annual frequency of testing (HbA1c, lipid profile, and urinary ACR) of each matched cohort is shown in Table 1, Table 2 and Table 3. In brief, the FM cohort had significantly more frequent HbA1c (3.2 vs. 2.8 times/year, *p* < 0.001) and lipid profile (7.3 vs. 7.0 times/year, *p* < 0.001) testing than the IM cohort. In contrast, the urinary ACR test was performed less frequently in the FM cohort than in the IM cohort (0.5 vs. 0.7%, *p* < 0.001). On average, the endocrinologist cohort checked the HbA1c 4.9 times, the lipid profile 10.6 times, and the ACR test 1.3 times annually, which were significantly more frequent than in either the FM or IM cohort (*p* < 0.001).

### 3.3. Total Diabetes-Related Costs

Compared with the endocrinologist cohort, the IM cohort had lower diabetes-related expenditure during the follow-up period (USD 502.05 vs. USD 932.27, *p* < 0.001). Moreover, the FM cohort had lower costs than the IM cohort (USD 487.41 vs. USD 507.67, *p* = 0.01) and only less than half of the total diabetes-related costs of patients in the endocrinology cohort (USD 484.39 vs. USD 927.85, *p* < 0.001) (Table 1, Table 2 and Table 3).

### 3.4. Prediction of the Occurrence of Acute Complications

Acute complications occurred significantly more frequently in the IM cohort than in the FM cohort (sHR (95% CI), 1.26 (1.08–1.47); *p* = 0.003) or endocrinologist cohort (sHR (95% CI), 1.10 (1.00–1.21), *p* = 0.046). The FM cohort had fewer complications compared to the endocrinologist cohort (sHR (95% CI), 1.57 (1.38–1.78), *p* < 0.001) and the least complications overall. Other important risk factors for developing acute complications were the urbanization level and presence of comorbidities. Patients living in low urbanization areas had higher risk of acute complications (1.40 to 1.45 folds) compared to those living in high urbanization areas. Comorbidities that influenced the risk of developing acute complications in all three cohorts were congestive heart failure (1.48 to 1.80 folds), peripheral vascular disease (1.56 to 1.94 folds), and cerebral vascular disease (1.64 to 1.79 folds). The testing frequency of HbA1c and ACR, frequency of insulin prescription, and diabetes-related costs were significantly but minimally related to the occurrence of acute complications (Table 4).

### 3.5. Incidence of Acute Complications

In the FM cohort and IM cohorts, 1.42% and 2.35% of patients were diagnosed with acute complications during the 2-year follow-up period, respectively (*p* < 0.001) (Table 1). We compared the cumulative incidence risks of acute complications in the FM and IM cohorts (Figure 2A), and the trends revealed that the cumulative incidence risk of developing acute complications in the IM cohort was greater than in the FM cohort over time (*p* < 0.001). When comparing the family medicine (FM) cohort to the endocrinologist cohort, acute complications occurred in 1.29% of patients in the FM cohort and 2.05% of patients in the endocrinologist cohort (*p* < 0.001) (Table 2). Furthermore, the trends of cumulative incidence of acute complications in the FM cohort were lower than those in the endocrinologist cohort over time (*p* < 0.001) (Figure 2B). We also compared the endocrinologist cohort with the internal medicine (IM) cohort, which showed a lower incidence of acute complication rate than the IM cohort (2.11% vs. 2.14%, *p* = 0.779) (Table 3). Trends revealed that the cumulative incidence of risks of developing acute complications in the endocrinologist cohort seemed greater initially but became lower than the IM cohort over time (*p* = 0.54) (Figure 2C).

## 4. Discussion

This analysis of 206,819 patients aged 18–75 years with newly diagnosed type 2 diabetes identified from the HWDC revealed that patients cared for by family physicians had fewer acute complications from diabetes than those cared for by endocrinologists or other internists. Our study also suggests that patients who were cared for by family physicians were more frequently given HbA1c and lipid profile testing than other internists, except endocrinologists. Lastly, we found that treatment by family physicians incurred lower health expenditures compared to endocrinologists and other internists, which makes the role of family physicians significant when striving for a cost-saving model of care.

On the basis of a previous study [10], the main problems affecting the quality of diabetes care include a disease-centered medical approach, lack of knowledge, and limited patient participation in decision-making. Family medicine residency training emphasizes comprehensive and continuous care for the individual. In addition to treating disease, family physicians focus on whole-person care and disease prevention such as smoking cessation and weight control. It is possible that the distinctive characteristics of family physicians may be responsible for the difference in outcomes in our study.

According to the 2021 American Diabetes Association guideline, [23] the standard of diabetes care includes annual checkups of HbA1c (4 times/year), LDL cholesterol (1 time/year), nephropathy (1 time/year), and eyes (1 time/year). We evaluated diabetes care among different specialties using these benchmarks, except for completion of the eye exam since data regarding this could not be reliably extracted from the existing information. It is noteworthy that all three cohorts monitored these process measures to a satisfactory degree, except for the screening of nephropathy in the FM and IM cohorts. However, a possible explanation for the lack of adequate monitoring in these two cohorts is that in Taiwan 65.1% of the diabetes patients exhibited doctor shopping behavior, 50.3% had no continuity of care, and 76.8% had no regular source of care [24]. Overall, the results warrant increased education for family physicians and internists to regularly monitor for the development of diabetic nephropathy.

Regarding cost, care by family physicians resulted in lower health expenditure for diabetes patients. The diabetes-related health expenditure includes payments from inpatient care, outpatient care, and prescriptions. However, these costs could be affected by the prescription of newer medicines, testing frequencies, and comorbidity-related care. When examining testing frequencies of standard diabetes care, [22] we found all cohorts received HbA1c tests and lipid panel examinations. Nevertheless, patients treated by endocrinologists had a higher frequency of HbA1c, lipid profile, and ACR testing than the FM cohort. The initiation of insulin was also the least frequent in the FM group compared to the other groups.

To the best of our knowledge, this is the first study to analyze the quality of diabetes care by examining both process and outcome indicators in patients treated by family physicians, endocrinologists, and other internists in Taiwan. This study has some limitations. First, miscoding of patient diagnoses by physicians in the HDWC system is possible. To mitigate such a bias, we composed an alternate definition based on previous studies to more accurately identify patients in the system with diabetes and is as follows: prescription for diabetes medication in the current year, and/or at least two diabetes codes from inpatient and/or outpatient visits over a 24 month period [17], which has been shown to have high sensitivity (93%) and specificity (98%). Additionally, Lin et al. [18] reported that identifying patients that have four or more outpatient visits or have one or more hospitalizations with a diabetes diagnosis code results in a higher accuracy of diabetes diagnosis. Instead of examining the outcomes of chronic complications, we only used acute complications as primary endpoints. Further research on chronic complications and long-term mortality outcomes is needed to create a complete picture of quality of diabetes care by different physicians. Thirdly, we could not obtain the initial HbA1c values and information regarding ophthalmologic exams from the HWDC. Therefore, we relied on other process indicators to reflect quality. Moreover, although we used the Charlson comorbidity index to match pairs of the three cohorts, there could be residual confounding beyond the CCI. For example, patients with comorbidities or risk factors unlisted in the CCI may not have the complexities of their situations properly reflected. Finally, the inclusion of only Taiwanese people in our study restricts the generalizability of our findings to other ethnic groups.

In this retrospective population-based cohort study comparing the care of newly diagnosed type 2 diabetes patients by family physicians, endocrinologists, and other internists, we found that family physicians provided favorable prevention of acute complications from diabetes at a lower cost. Overall, our study is useful for determining cost-effectiveness and informing policy efforts at improving diabetes care. However, these results could be due to the large sample size and need to be validated in future studies.

## Figures and Tables

**Figure 1 jpm-12-00461-f001:**
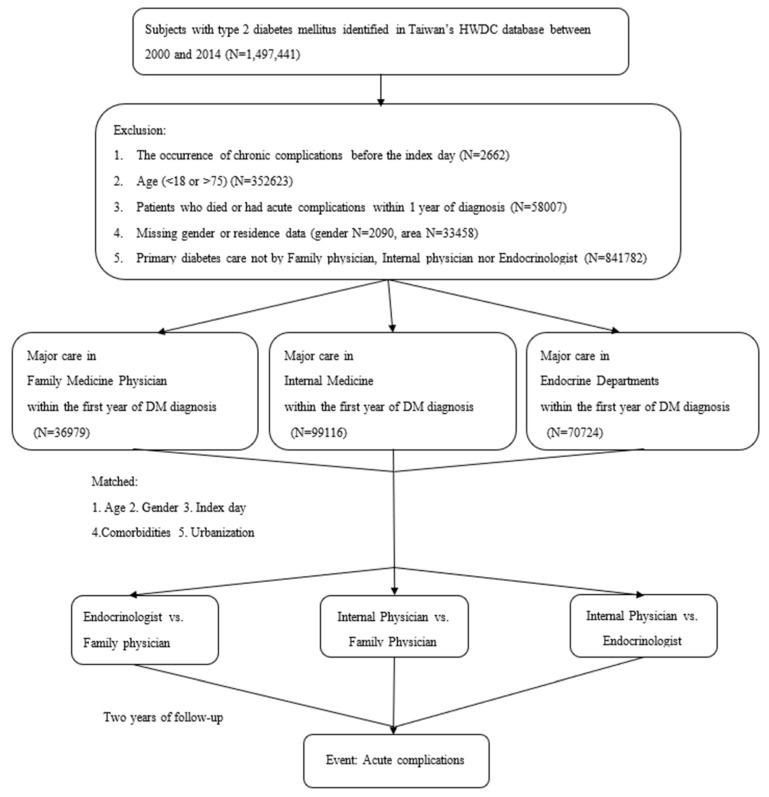
Study flow. Enrollment, distribution of each stage, and the end of the follow-up process for study eligible participants in the Health and Welfare Data Science Center (HWDC).

**Figure 2 jpm-12-00461-f002:**
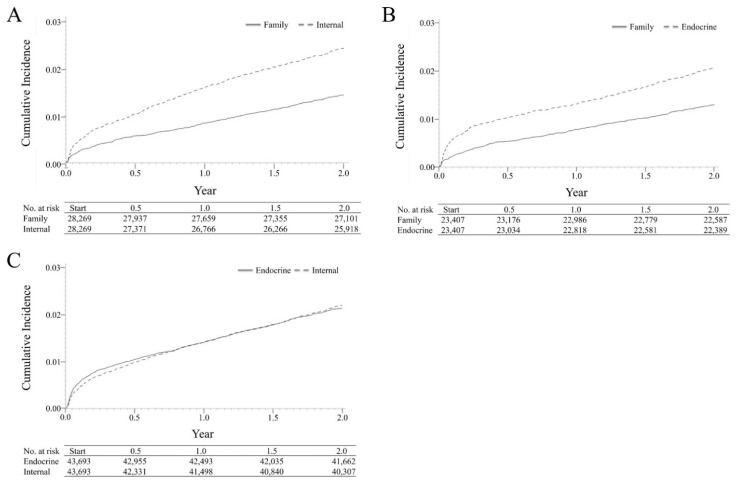
Cumulative Incidences of Acute Complications. (**A**) Family medicine and other internal medicine cohorts. (**B**) Family medicine and endocrinologist cohorts. (**C**) Other internal medicine and endocrinologist cohorts. The survival curves of cumulative incidence risks of acute complications for the FM, IM, and endocrinology cohorts were compared in pairs using the log-rank test. (**A**) Trends revealed that the cumulative incidence risks of developing acute complications in the IM cohort were greater than those in the FM cohort over time. (**B**) Cumulative incidences of acute complications in the FM cohort were lower than those in the endocrinologist cohort over time. (**C**) Trends revealed that the cumulative incidence of risks of developing acute complications in the endocrinologist cohort seemed greater initially but became lower than that in the IM cohort eventually.

**Table 1 jpm-12-00461-t001:** Characteristics of the family medicine (FM) and other internal medicine (IM) cohorts ^a^.

	FM *n* = 28,269	IM *n* = 28,269	*p*-Value
Age	56.43 ± 10.03	54.99 ± 10.22	<0.001
Gender—no. (%)			>0.999
Female	15,940 (56.39)	15,940 (56.39)	
Male	12,329 (43.61)	12,329 (43.61)	
Urbanization—no. (%)			>0.999
High	11,309 (40.00)	11,309 (40.00)	
Moderate	12,454 (44.06)	12,454 (44.06)	
Low	4506 (15.94)	4506 (15.94)	
Testing frequency (per year)			
HbA1C	3.19 ± 2.75	2.8 ± 2.84	<0.001
Lipid profile	7.29 ± 7.09	6.97 ± 7.41	<0.001
ACR	0.46 ± 1.13	0.68 ± 1.74	<0.001
Insulin (1 month) ^b^	0.3 ± 1.42	0.45 ± 1.78	<0.001
Cost ^c^	487.41 ± 590.97	507.67 ± 853.62	0.01
Comorbidities—no. (%)			
Myocardial infarct	875 (3.1)	912 (3.23)	0.374
Congestive heart failure	1646 (5.82)	1692 (5.99)	0.412
Peripheral vascular disease	1188 (4.2)	1227 (4.34)	0.417
Cerebrovascular disease	2882 (10.19)	2820 (9.98)	0.387
Dementia	47 (0.17)	40 (0.14)	0.453
Chronic lung disease	5401 (19.11)	5343 (18.9)	0.534
Connective tissue disease	221 (0.78)	203 (0.72)	0.380
Ulcer	8469 (29.96)	8431 (29.82)	0.727
Chronic liver disease	7726 (27.33)	7974 (28.21)	0.02
Hemiplegia	15 (0.05)	10 (0.04)	0.317
Moderate or severe kidney disease	701 (2.48)	688 (2.43)	0.724
Tumor, leukemia, lymphoma	2537 (8.97)	2518 (8.91)	0.779
Moderate or severe liver disease	137 (0.48)	138 (0.49)	0.952
Malignant tumor, metastasis	537 (1.90)	555 (1.96)	0.582
AIDS	0	0	
Acute Complications—no. (%)	401 (1.42)	665 (2.35)	<0.001

^a^ Plus–minus values are means ± SD. ^b^ Insulin (1 month) means the initial prescription frequencies during the first month of diagnoses. ^c^ Cost means the average diabetes-related annual costs expressed in USD. ACR = albumin-to-creatinine ratio. AIDS = acquired immunodeficiency syndrome. HbA1c = glycated hemoglobin.

**Table 2 jpm-12-00461-t002:** Characteristics of the family medicine (FM) and endocrinologist cohorts ^a^.

	FM *n* = 23,407	Endocrinologist *n* = 23,407	*p*-Value
Age	55.48 ± 10.09	54.54 ± 10.11	<0.001
Gender—no. (%)			>0.999
Female	13,274 (56.71)	13,274 (56.71)	
Male	10,133 (43.29)	10,133 (43.29)	
Urbanization—no. (%)			>0.999
High	10,455 (44.67)	10,455 (44.67)	
Moderate	9974 (42.61)	9974 (42.61)	
Low	2978 (12.72)	2978 (12.72)	
Testing frequency (per year)			
HbA1C	3.23 ± 2.76	4.89 ± 3.52	<0.001
Lipid profile	7.42 ± 7.13	10.58 ± 8.34	<0.001
ACR	0.46 ± 1.13	1.27 ± 1.93	<0.001
Insulin (1 month) ^b^	0.28 ± 1.28	0.81 ± 2.44	<0.001
Cost ^c^	484.39 ± 574.26	927.85 ± 922.90	<0.001
Comorbidities—no. (%)			
Myocardial infarct	626 (2.67)	647 (2.76)	0.551
Congestive heart failure	1181 (5.05)	1295 (5.53)	0.019
Peripheral vascular disease	633 (2.7)	684 (2.92)	0.154
Cerebrovascular disease	2223 (9.5)	2387 (10.2)	0.011
Dementia	48 (0.21)	54 (0.23)	0.552
Chronic lung disease	3816 (16.3)	3876 (16.56)	0.454
Connective tissue disease	269 (1.15)	287 (1.23)	0.443
Ulcer	6038 (25.8)	6106 (26.09)	0.473
Chronic liver disease	5225 (22.32)	5077 (21.69)	0.099
Hemiplegia	3 (0.01)	3 (0.01)	>0.999
Moderate or severe kidney disease	584 (2.49)	523 (2.23)	0.064
Tumor, leukemia, lymphoma	1949 (8.33)	2023 (8.64)	0.220
Moderate or severe liver disease	50 (0.21)	50 (0.21)	>0.999
Malignant tumor, metastasis	212 (0.91)	216 (0.92)	0.846
AIDS	0	0	
Acute Complications—no. (%)	302 (1.29)	479 (2.05)	<0.001

^a^ Plus–minus values are means ± SD. ^b^ Insulin (1 month) means the initial prescription frequencies during the first month of diagnoses. ^c^ Cost means the average diabetes-related annual costs expressed in USD. ACR = albumin-to-creatinine ratio. AIDS = acquired immunodeficiency syndrome. HbA1c = glycated hemoglobin.

**Table 3 jpm-12-00461-t003:** Characteristics of the endocrinologist and other internal medicine (IM) cohorts ^a^.

	Endocrinologist *n* = 43,693	IM *n* = 43,693	*p*-Value
Age	54.08 ± 10.56	53.59 ± 10.73	<0.001
Gender—no. (%)			>0.999
Female	25,508 (58.38)	25,508 (58.38)	
Male	18,185 (41.62)	18,185 (41.62)	
Urbanization—no. (%)			>0.999
High	22,467 (51.42)	22,467 (51.42)	
Moderate	16,974 (38.85)	16,974 (38.85)	
Low	4252 (9.73)	4252 (9.73)	
Testing frequency (per year)			
HbA1C	4.95 ± 3.61	2.84 ± 2.87	<0.001
Lipid profile	10.69 ± 8.48	7.15 ± 7.53	<0.001
ACR	1.25 ± 1.89	0.68 ± 1.71	<0.001
Insulin (1 month) ^b^	0.83 ± 2.43	0.43 ± 1.73	<0.001
Cost ^c^	932.27 ± 943.78	502.05 ± 839.06	<0.001
Comorbidities—no. (%)			
Myocardial infarct	806 (1.84)	825 (1.89)	0.635
Congestive heart failure	2454 (5.62)	2442 (5.59)	0.860
Peripheral vascular disease	1688 (3.86)	1636 (3.74)	0.358
Cerebrovascular disease	4573 (10.47)	4271 (9.78)	<0.001
Dementia	94 (0.22)	75 (0.17)	0.144
Chronic lung disease	7055 (16.15)	6744 (15.43)	0.004
Connective tissue disease	269 (0.62)	258 (0.59)	0.631
Ulcer	12,006 (27.48)	11,980 (27.42)	0.844
Chronic liver disease	10,059 (23.02)	10,345 (23.68)	0.022
Hemiplegia	56 (0.13)	47 (0.11)	0.375
Moderate or severe kidney disease	831 (1.90)	986 (2.26)	<0.001
Tumor, leukemia, lymphoma	3420 (7.83)	3443 (7.88)	0.772
Moderate or severe liver disease	373 (0.85)	392 (0.90)	0.490
Malignant tumor, metastasis	597 (1.37)	609 (1.39)	0.728
AIDS	0	0	
Acute Complications—no. (%)	924 (2.11)	936 (2.14)	0.779

^a^ Plus–minus values are means ± SD. ^b^ Insulin (1 month) means the initial prescription frequencies during the first month of diagnoses. ^c^ Cost means the average diabetes-related annual costs expressed in USD. ACR = albumin-to-creatinine ratio. AIDS = acquired immunodeficiency syndrome. HbA1c = glycated hemoglobin.

**Table 4 jpm-12-00461-t004:** Prediction of the occurrence of acute complications ^a^.

	Endocrine vs. FM		IM vs. FM		IM vs. Endocrine	
	sHR ^d^ (95% CI)	*p*-Value	sHR (95% CI)	*p*-Value	sHR (95% CI)	*p*-Value
Endocrine vs. FM	1.57 (1.38–1.78)	<0.001				
IM vs. FM			1.26 (1.08–1.47)	0.003		
IM vs. Endocrine					1.10 (1.00–1.21)	0.046
Age	1.00 (0.99–1.01)	0.780	1.00 (1.00–1.01)	0.210	1.00 (0.99–1.00)	0.669
Male vs. Female	1.06 (0.93–1.20)	0.383	1.01 (0.87–1.16)	0.942	0.99 (0.90–1.09)	0.784
Urbanization						
High	REF.		REF.		REF.	
Moderate	1.22 (1.06–1.39)	0.006	0.99 (0.84–1.15)	0.869	1.19 (1.08–1.31)	<0.001
Low	1.45 (1.22–1.72)	<0.001	1.41 (1.15–1.72)	<0.001	1.40 (1.21–1.63)	<0.001
Testing frequency (per year)						
HbA1C	0.94 (0.91–0.97)	<0.001	0.97 (0.95–1.00)	0.091	0.98 (0.96–0.99)	0.01
Lipid profile	1.01 (1.00–1.02)	0.140	1.01 (0.99–1.02)	0.267	1.00 (0.99–1.01)	0.832
ACR	1.08 (1.06–1.09)	<0.001	1.03 (0.99–1.07)	0.133	1.05 (1.03–1.07)	<0.001
Insulin (1 month) ^b^	1.07 (1.05–1.09)	<0.001	1.07 (1.05–1.08)	<0.001	1.07 (1.05–1.08)	<0.001
Cost ^c^	1.00 (1.00–1.00)	<0.001	1.00 (1.00–1.00)	<0.001	1.00 (1.00–1.00)	<0.001
Comorbidities						
Myocardial infarct	1.62 (1.27–2.07)	<0.001	1.36 (0.99–1.88)	0.061	1.53 (1.22–1.93)	<0.001
Congestive heart failure	1.80 (1.48–2.19)	<0.001	1.48 (1.15–1.89)	0.002	1.68 (1.44–1.97)	<0.001
Peripheral vascular disease	1.85 (1.50–2.29)	<0.001	1.56 (1.16–2.10)	0.004	1.94 (1.65–2.29)	<0.001
Cerebrovascular disease	1.72 (1.46–2.02)	<0.001	1.79 (1.48–2.15)	<0.001	1.64 (1.45–1.86)	<0.001
Dementia	2.00 (0.64–6.22)	0.233	0.51 (0.07–3.67)	0.508	1.35 (0.56–3.26)	0.501
Chronic lung disease	0.91 (0.77–1.06)	0.227	0.82 (0.67–1.01)	0.065	0.96 (0.85–1.10)	0.570
Connective tissue disease	1.63 (0.99–2.69)	0.055	1.51 (0.89–2.57)	0.126	1.46 (0.93–2.30)	0.101
Ulcer	0.98 (0.85–1.12)	0.720	0.94 (0.79–1.11)	0.440	1.04 (0.94–1.16)	0.439
Chronic liver disease	1.08 (0.94–1.24)	0.289	0.99 (0.83–1.19)	0.913	0.99 (0.89–1.12)	0.914
Hemiplegia	1.05 (0.15–7.52)	0.962	-		0.65 (0.16–2.64)	0.549
Moderate or severe kidney disease	1.29 (0.93–1.80)	0.133	1.38 (0.94–2.02)	0.096	1.09 (0.82–1.45)	0.569
Tumor, leukemia, lymphoma	0.84 (0.64–1.10)	0.200	1.01 (0.76–1.34)	0.965	0.81 (0.66–1.00)	0.048
Moderate or severe liver disease	1.04 (0.43–2.52)	0.936	0.67 (0.09–4.78)	0.687	2.23 (1.53–3.24)	<0.001
Malignant tumor, metastasis	1.44 (0.86–2.41)	0.165	1.71 (0.88–3.33)	0.115	1.99 (1.34–2.97)	<0.001
AIDS	-		-		-	

^a^ Means not applicable. ^b^ Insulin (1 month) means the initial prescription frequencies during the first month of diagnoses. ^c^ Cost means the average diabetes-related annual costs expressed in USD. ^d^ The subdistribution hazard ratio (sHR) calculated using the Fine and Gray regression hazards model, and *p*-values were determined using Gray’s test. ACR = albumin-to-creatinine ratio. AIDS = acquired immunodeficiency syndrome. Endocrine = the endocrinologist cohort. FM = the family medicine cohort. HbA1c = glycated hemoglobin. IM = the other internal medicine cohort.

## Data Availability

Restrictions apply to the availability of these data. Data was obtained from HWDC and are available with the permission of HWDC.

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
