# Peer review of "Newly Diagnosed Type 2 Diabetes Care between Family Physicians, Endocrinologists, and Other Internists in Taiwan: A Retrospective Population-Based Cohort Study"

_jpm, 2022, doi:10.3390/jpm12030461_

Round 1

Reviewer 1 Report

Table 4: the value for Hemoplegia under the comorbidities section is missing for IM vs FM cohort?

Author Response

Thanks for your time to review our manuscript again! We deeply appreciate it. 

We have marked the value of Hemiplegia in Table 4 as "-" which stands for "Not Applicable" when comparing IM yo FM cohorts. 

Reviewer 2 Report

The authors addressed all my comments for the previous submission

Author Response

Thanks for your time to  our manuscript again! We deeply appreciate it.

This manuscript is a resubmission of an earlier submission. The following is a list of the peer review reports and author responses from that submission.

Round 1

Reviewer 1 Report

In this manuscript, Chou and colleagues compared the incidence of acute complications in patients with newly diagnosed type 2 diabetes from three different care settings, i.e. family medicine, endocrinologist, and other internal medicine physicians. They found that patients followed in the primary care setting have a lower incidence of complications and generate a lower burden of costs. The study is well performed, the size of the population is huge, and the limitations of the study are well stated and addressed.

My only concern is that the authors do not provide information on the criteria for the patients to be assigned to a specific care setting. Are the three patients group homogenous in terms of baseline HbA1c, comorbidities, and other risk factors? This may explain the different incidence of complications among groups. Indeed, in a real-life setting, a newly diagnosed T2DM patient may be assigned to a specific setting according to the probability of developing complications.

Moreover, the authors should considerably improve the quality of figures, and merge all the panels of figure 2 into a single image.

Reviewer 2 Report

Short summary: 

Pei-lin Chou et al. conducted a retrospective population-based cohort study to compare the quality and cost of services received by newly diagnosed type 2 diabetes patients. Specifically, process indicators, health expenditure on diabetes care, and incidence of acute complications are compared among those patients who received their primary care from family physicians, endocrinologists, or other internists. Since diabetes has become one of the most common diseases with a huge social burden, the question that the authors aimed to answer has big impacts on both medical and social aspects. With the large number of patients involved in this study, the research results are more realistic and reliable.

General comments:

  1. More background and introduction are needed to support the study design. In the introduction section, the authors mentioned that “a long-standing debate” on whether treatment by different specialists leads to a different quality of diabetes care”. How to measure the quality of diabetes care? I believe the quality of diabetes care can be measured in multiple aspects, what are they? Are there any existing criteria to measure the quality of diabetes care? Will the “key outcomes” used in this manuscript represent the quality measurement criteria? More background introduction and reference information are needed on this topic. Few examples:
    1. The frequency of insulin usage is defined as the initial prescription frequencies during the first month. The first month of what? The first month of diagnosis or first month of insulin usage, or another timeframe? The prescription frequency is associated with the severity of diabetes. Can the prescription frequencies represent insulin usage? How about the prescription dosage? Is it better to use long-term insulin usage to reflect the control of diabetes?
    2. Why is ACR selected as one process indicator? ACR is used to evaluate nephropathy, why not other exams? How will the severity of diabetes affect the test frequency? If the majority of patients have no nephropathy, the conclusion built based on the ACR frequency will be largely biased by a small group of patients who developed nephropathy.
  2. In this whole study, only limited information is collected for each patient, such as age, sex, level of urbanization, etc. I understand due to the data limitation, it is a trade-off between large population cohorts and completeness of data. But some critical information are not collected and controlled in the study which can affect the conclusions authors made. Few examples:
    1. Socioeconomic status is not available but will have a large impact on the analysis results and conclusions. Patients with higher economic status may prefer more expensive imported drugs, willing to do more tests or have access to high-end hospitals which all will impact the cost of care. Patients with lower economic levels or from rural areas may not have access to an endocrinologist but a family physician or have difficulty adhering to doctors’ orders, such as lifestyle change, insulin usage, etc., which can lead to a higher incidence of acute compliances.
    2. Most newly diagnosed patients haven’t developed comorbidities yet. Therefore, the severity of newly diagnosed type 2 diabetes is critical. It can change the primary care physician patients select, insulin usage, long-term diabetes control, acute complications, etc. 
    3. Though the level of urbanization information is collected, but not controlled between the comparing cohorts. As we can see from the results table, there exist big differences in urbanization levels among the three cohorts. The urbanization level may limit the access to specialists and tests, cost of the services, etc.
    4. In the Materials and methods or supplementary, it is worth describing how the data quality control was achieved, any procedures for data cleaning purposes.
  3. There exist some study limitations but not well addressed by the authors. Few examples:
    1. All enrolled patients were followed for 2 years to calculate the incidence of acute complications, but the definition used to split patients into three cohorts is 1-year visit information after the diagnosis. For example, Will the patients switch their primary care physician in the second year? The switch can be caused by many reasons, such as change of diabetes’ severity, location of living, etc. Those reasons may have a big impact on the conclusions.
    2. Tables 1, 2, 3, age and age group comparisons show significant differences (small p-value) between the comparing two cohorts. It is reasonable to suspect that this significant difference is caused by the large sample size. However, if this assumption holds, the differences we observed for other factors, such as the cost between FM and IM, could also be caused by the large sample size. There is no good way to avoid it, but at least it needs to be discussed.
    3. Patients who did not stick with one type of physician were excluded, will this group of patients have certain patterns which are not considered? 
  4. Description inconsistency. For example:
    1. Enrolled patients are newly diagnosed with type 2 diabetes. But this definition is not well kept through the manuscript.
    2. In the study design and study population, “including patients aged 18-75 … between 2000 and 2014”. But in Figure A. “subjects with type 2 diabetes mellitus were … between 2001 and 2015”. 2014 or 2015? Subject with type 2 diabetes or subject newly diagnosed with type 2 diabetes?
  5. There are grammar errors and unclear descriptions. Some paragraphs need to rearrange the sentence order for a better logical flow. 

Specific comments:

  1. Abstract, background. “...outcomes, costs, and process of care”. Care of what? Need to specify the topic clearly.
  2. Abstract, Methods. “... were classified into three cohorts: …”, patients were classified into three cohorts based on what? “Based on their primary care physician type”? Need to be clear. 
  3. Introduction; 2nd paragraph. “while the knowledge base of endocrinologists is strong, the general …. approach”. 

Please provide a reference for this conclusion if available.

  1. Introduction; 2nd paragraph. “There has been a long-standing debate on whether treatment by … diabetes care”. 

It would be interesting if the authors can provide more background and references about the “quality of diabetes care”. How do we evaluate the “quality of diabetes care”, are there any existing criteria? This discussion will help to strengthen the choices why process indicators, health expenditure on diabetes care, and incidence of acute complications are used as the “key outcomes''? 

  1. Materials and methods - study design and study population, 1st paragraph. 
    1. “... between 2000 and 2014, and collected age (18-39/40-59/60-75)...” Why were patients who are older than 75 were excluded? Any specific reasons? Based on Tables 1, 2, 3, the numeric value of age was also collected, not just the age group. These details must be clearly and consistently described.
    2. “... and comorbidities that existed prior to the index day in the Charlson comorbidity index … at baseline (i.e. the index date)”. What is the “index day”? Should be defined here rather than later. What is the “baseline”? What is the “index date”? The timeline of patients enrollment and follow-up is not clear. It may be better to have a patient-level timeline figure.
    3. “...not incident type 2 diabetes… with any DM diagnosis…”. DM is not defined.
  2. Tables 1, 2, 3. What is the number before and in the brackets? What is the value before and after the plus and minus? Though readers can make a correct guess, it must be clearly defined, like what authors have in table 4.